# Feral Donkey Distribution and Ecological Impacts in a Hyper-Arid Region

**DOI:** 10.3390/ani13182885

**Published:** 2023-09-11

**Authors:** Alaaeldin Soultan, Mohammed Darwish, Nawaf Al-Johani, Ayman Abdulkareem, Yousef Alfaifi, Abdulaziz M. Assaeed, Magdy El-Bana, Stephen Browne

**Affiliations:** 1The Royal Commission for AlUla, Riyadh 12512, Saudi Arabia; m.darwish@rcu.gov.sa (M.D.); n.aljohani@rcu.gov.sa (N.A.-J.);; 2Plant Production Department, College of Food & Agriculture Sciences, King Saud University, Riyadh 11451, Saudi Arabia; 3Department of Botany, Faculty of Science, Port Said University, Port Said 42511, Egypt

**Keywords:** feral donkey, Arabian Desert, AlUla, overgrazing, density

## Abstract

**Simple Summary:**

We estimated the population density and the population size of feral donkeys in northwest Saudi Arabia. The estimated overall population density was 1.03 (0.19 SE) donkey/km^2^ and an abundance at ~1135 individuals. The negative impact of feral donkeys on natural resources included overgrazing, habitat fragmentation, and competition for resources with native species. Our study adds evidence to the detrimental impacts of feral donkeys and calls for urgent actions to control the presence of feral donkeys in the region. We recommend humane eradication for controlling the feral donkeys as it would be the most efficient and applicable action that can significantly abate the feral donkeys’ negative impacts.

**Abstract:**

The feral donkey (*Equus asinus* L.) is an invasive species in Saudi Arabia and can cause severe damage to natural and cultural heritage. Over the last 30 years, feral donkeys have become a serious problem, as their abundance and geographic distribution has increased drastically. The impacts of feral donkeys are not well documented, and information about their abundance and distribution is lacking, certainly in Saudi Arabia, which hampers the implementation of effective management plans. Accordingly, we used the minimum population number approach (MPN) to determine the number of feral donkeys in this part of northwest Saudi Arabia. A total of 1135 feral donkeys were encountered in the region. The area around Khaybar harbors ~25% (*n* = 338) of the feral donkey population, whereas Tayma and AlGhrameel nature reserves were the least-inhabited sites (almost absent). The average population density of feral donkeys was estimated as 1.03 (0.19 SE) donkey/km^2^. We documented the negative ecological impact of feral donkeys on natural resources, which constituted overgrazing that resulted in habitat fragmentation and competition for resources with native species. We propose urgent actions to control the presence of feral donkeys in the region and suggest humane eradication as the most efficient and applicable to significantly reduce the negative impacts of feral donkeys.

## 1. Introduction

For more than 4000 years, donkeys have been domesticated as beasts of burden for packing, transporting goods, and tillage [1]. Donkeys (*Equus asinus* L.), also called “feral burros” [2], are believed to be evolved from the African wild ass (*Equss africanus*) and lived and worked with humans for 1000s of years in Arabia [3]. They are an equine species with distinct large heads and long ears [4]. The long ears allow donkeys to hear distant calls of other donkeys and are used for cooling in hot weather [5]. They vary in color from white to black, while dun grey is the most common color pattern [4]. Most often, feral donkeys have dorsal and shoulder strips that are primitive markings [6]. Donkeys have a specialized digestive system that allows the consumption of large amounts of low-quality forage and the subsequent extraction of nutrients [2]. Donkeys’ weight and height range between 110 and 142 kg and between 102 and 142 cm, respectively [6]. These variations are a result of the quantity and quality of the available food [7,8]. Donkeys were used for their strength, agility, and adeptness in navigating terrain that was unsuitable for both horses and camels [3]. Since the 1970s, the establishment of an extensive road network has reduced the need for donkeys, and they were set free into the wild, to become feral and now breed and graze in unmanaged herds [9].

The IUCN Invasive Species Specialist Group (ISSG) classified feral donkeys in Saudi Arabia as an invasive species in 2011 [10]. The FAO Statistics Division estimated a population of 55,000 donkeys in 1961 and 100,000 by 2020 in Saudi Arabia [11]. The population size of feral donkeys presents significant ecological problems with consequential socioeconomic implications. For instance, previous studies documented the negative impact of donkey grazing behavior on the availability of forage for other herbivores [12,13,14]. Further, the presence of feral donkeys has various effects on native wildlife, including competition for food and shelter, habitat destruction, and disease transmission [15]. In arid regions, feral equines can spend up to two-thirds of their day at water sources, using aggressive behavior to prevent native wildlife from using these resources [16,17].

Despite the negative ecological impacts of feral donkeys on ecosystem processes and functions, their associated impacts have not been well documented in the Arabian Desert and particularly in Saudi Arabia. Moreover, the numbers of feral donkeys and their distribution are still unknown and no measures have yet been taken to reduce their impact.

The Royal Commission for AlUla (RCU) has committed to restoring and conserving the natural heritage of the Arabian Desert, specifically in the AlUla region. To this end, RCU has established five nature reserves in the region and implemented several restoration and reintroduction projects within these reserves. However, there was a concern that the presence of feral animals, particularly donkeys, could challenge the success of these projects. Accordingly, it is important for the RCU to determine the current spatial distribution and the relative abundance of the feral donkeys, and to evaluate their associated impacts on the natural resources. Such information can be used to inform conservation managers to implement an effective control plan for feral donkeys. Therefore, this study was aimed at assessing the current distribution of feral donkeys and estimating their population size in the region. To this end, we developed a stratified sampling design to estimate the population size of feral donkeys and model the association between the estimated number and the environmental variables.

## 2. Materials and Methods

### 2.1. Study Area

The study area is located in the northwest of Saudi Arabia (Figure 1) and under the authority of the Royal Commission for AlUla (RCU). The study area covers a total area of ~25,000 km^2^ that enclosed AlUla County and the surrounding nature reserves, AlGhrameel Nature Reserve (GNR), Harrat Uwayrid Biosphere Reserve (HUBR), Sharaan Nature Reserve (SNR), Harrat AlZabin Nature Reserve (HZNR), and Wadi Nakhlah Nature Reserve (WNNR). Three other areas were also covered during the survey, Jabal ElWard (120 km west of AlUla County), Tayma (70 km northeast of AlUla County), and Khaybar (200 km southeast of AlUla County) (Figure 1). All reserves are unfenced, with the exception of SNR, which was not included in the survey. The study area is an inland desert habitat that varies in altitude (200–2000 m.a.s.l.). It is characterized by heterogeneous habitats of lava fields, plateaus, sandstone outcrops, sand dunes, and several permanent springs that support wetland and oasis habitats. The RCU’s nature reserves were designed to represent key desert habitats and to facilitate structural and functional connectivity with other reserves. For instance, King Salman bin Abdulaziz Royal Natural Reserve is connected with RCU’s nature reserves through GNR and Tayma, while Prince Mohammed bin Salman Royal Reserve is connected with RCU’s nature reserves through HUBR.

### 2.2. Data Collection and Analysis

Initially, we developed a stratified sampling design to estimate the population size of feral donkeys within the study area, based on a distance sampling approach [18]. However, applying this approach was very challenging, due to the donkey’s behavior and uneven use of the study area. Feral donkeys are highly sociable and form large groups of around 15 individuals [19]. When humans are present, they emit loud braying sounds as a warning to others, prompting them to flee and disappear into the surroundings [20]. This behavior poses a challenge to distance sampling assumptions, which rely on the detection of objects at their original location and precise measurements [18]. As a result, estimating the detection probability of this species becomes difficult. Additionally, since this study is a baseline to inform the feral animal control plan, rather than a population estimate for long-term monitoring, we decided to use an alternative simpler approach. Hence, we used the minimum population number approach (MPN) to determine the number of feral donkeys in the study area [14,21]. A recent study showed that MPN can calculate the population number with high accuracy and precision and also facilitates temporal comparison [21]. We divided the study area including Jabal ElWard into 900 grids, 5 km × 5 km each. A total of 109 grids were selected to represent the study area. Since the distribution of feral donkeys was uneven across the landscape, the selection of these grids was based on the result of the interview with the local community and shepherds. Additionally, we considered the distance between the selected grids to minimize spatial autocorrelation and to ensure the independence of the observations. For Khaybar and Tayma, due to their small spatial scales (~50 and 20 km^2^, respectively), each was sampled by a single 5 km × 5 km grid. We used a daytime survey to collect field data from the grids and to calculate the MPN in the study area. The survey was carried out between February and May 2022. This involved both driving transect (~25 km/h) along dirt roads and walking transect through inaccessible areas (e.g., narrow valleys, outcropped habitats, wetlands, and abandoned farms). For the whole study area, excluding GNR and Tayma, transect design (length and the direction) was unsystematic and determined by the topographic feature of the grids, whereas in GNR and Tayma, with a dominant open habitat such as gravel plain and sand sheets, the grids were surveyed using two parallel transects (2 km apart) with a southeast–northwest orientation. The survey was carried out with a team of four observers. For each sighting of a feral donkey, the location (using Garmin GPSMAP 64s, Taiwan) and the number of animals were recorded. The MPN was calculated as the total number of all recorded animals across the study area, while feral donkey population density was calculated as the number of recorded donkeys/km^2^.

The landscape covariates associated with each grid were also recorded, which include human presence, vegetation cover, presence of water source (yes/no), and topographic roughness. Human presence was calculated as a perpendicular distance to the nearest settlement or farm. The vegetation cover was estimated visually along the sampled transects on a continuous scale between “0” (no vegetation) and “100” (complete cover). The topographic roughness was used here to represent the surface evenness and was calculated from the elevation data using the “terrain” function implemented in “raster” R package [22].

We used the Generalized Linear Mixed Models (GLMMs) to test whether the MPN of feral donkeys was driven by the associated landscape covariates [23]. GLMMs allow for fitting ecological data that are not normally distributed while accounting for random effects by including the sites as a random factor [24,25]. We fitted GLMMs with all possible combinations including interaction terms using the Poisson distribution using the package glmmTMB [26]. The best-fitted model was selected using Akaike’s Information Criterion (AIC) [27].

The spatial distribution of the feral donkeys was mapped using Inverse Distance Weighted interpolation (IDW), a deterministic spatial interpolation approach [28]. The fundamental principle behind IDW is that the distance between a known point and an unknown point influences the unknown property being estimated (i.e., the impact decreases as the distance increases) [28].

## 3. Results

A total survey effort of 2225 km covered the 109 grids across the study area. The calculated MPN derived was ~1135 feral donkeys (Table 1). The Khaybar area was the most inhabited site with 338 individuals, whereas Tayma and GNR were the least inhabited sites (almost absent). The average population density was estimated as 1.03 (0.19 SE) feral donkeys/km^2^. This density was estimated over the study area including Jabal ElWard, Tayma, and Khaybar.

The model that included both the linear and quadratic terms of the presence of human and vegetation cover covariates was the best supported (lowest AIC = 207.1) (Table 2). GLMM revealed an association between the MPN of feral donkeys and both the presence of humans (*z* = −3.690, *p* = 0.0001; Figure 2(a1)) and the extent of vegetation cover (*z* = −4.359, *p* = 0.0001; Figure 2(a2)). Although the best-fitted model showed no statistically significant impact of topographic roughness, feral donkeys were observed as tending to occupy sites that are dominated by rocky outcrop habitats and avoid open plains. The spatial distribution of the calculated MPN showed a high probability of occurrence for feral donkeys on the western side of the study area compared to the eastern side (Figure 2b).

## 4. Discussion

This study assesses the distribution and population size of the feral donkey in the AlUla region and shows that at a minimum there are approximately 1135 feral donkeys occurring in the study area. The average population density was estimated as 1.03 (0.19 SE) donkeys/km^2^, with a group size between 5 and 28 individuals. We acknowledge that our estimate was conservative and certainly underestimated the number of feral donkeys in the study area. Nevertheless, this underestimated figure strongly indicates that feral donkeys are overabundant and present a management issue.

Khaybar area harbors the highest number, ~338, of feral donkeys. This could be explained by the presence of ~350 natural water points and several abandoned farms, which allow survival in the extreme desert conditions. Conversely, the near absence of feral donkeys at GNR and Tayma could be attributed to water scarcity and the degradation of the vegetation cover (pers. obs.). Additionally, both GNR and Tayma are open habitats, mostly sand and gravel plains with no outcrops, which makes donkey survival in these areas difficult due to the direct exposure to harsher conditions. Despite these conditions, grazing is still possible in the area where the shepherds provide food (e.g., Alfalfa) and water for their flocks. Such practices might increase the possibility of feral donkeys occupying both GNR and Tayma in the very near future.

Our model shows a positive association between feral donkey population size and human activity. This finding is consistent with a previous study that found a high population density of donkeys in proximity to human presence in South Africa [14]. This is expected as donkeys are taking advantage of the resources provided by humans, such as fodder for their livestock. This association between feral donkeys and humans could increase their negative impacts, such as road traffic accidents and disease transmission, due to the continuous interaction with livestock. The best-fitted model showed no statistical significance for topography. Although this finding is consistent with a previous study [20], we observed a tendency for donkeys to dwell on site with relatively high topographic variability.

During our fieldwork, we recorded a multitude of adverse impacts for feral donkeys, including ecological and socio-economic impacts. Unlike native herbivore species such as Arabian oryx (*Oryx leucoryx*), Sand gazelle (*Gazella marica*), Arabian gazelle (*Gazella arabica*), and Nubian ibex (*Capra nubiana*), feral donkeys were observed to uproot native plant species during their grazing. Further, they also negatively affect the native plants through overgrazing, selective bark stripping of woody species, and trampling due to their tendency to form herds and their high population size. These detrimental effects of feral donkeys on floristic composition and diversity have been reported by other studies [9,29,30], and they can result in deterioration of ecosystem processes and functions of arid regions [31,32] and contribute to soil erosion [33].

In the deserts of North America, bark damage induced by feral donkeys resulted in significant deterioration in the recruitment and the reproduction of native trees [34]. In our study, feral donkeys were also observed to damage Acacia trees (*Vachellia* spp.; keystone species in the study area) by girdling the bark of the trees. This behavior could be a result of drought, lacking some minerals or salts in donkeys’ diets, and they might find moisture and minerals in Acacia trees’ bark. However, girdling was very common across the study area and would increase the possibility of Acacia dieback [34].

Living in large groups allows feral donkey to dominate available resources, and their aggressive behavior can drive native wildlife from an area [35]. For instance, feral donkeys were reported to pollute and foul water resources and prevent other native species from drinking [36]. A recent study showed that the presence of feral donkeys in the study area resulted in a ~50% decline in the number of the endemic Arabian Partridges *Alectoris melanocephala* [37]. In addition, feral donkeys can also facilitate disease transmission between wildlife and domestic animal [15,38].

Feral donkeys also have a socio-economic impact, as they do become associated with human settlements and are reported as destroying local people’s assets and properties. It is a very common behavior now for donkeys to approach human settlements and overturn garbage bins looking for food, which, in turn, increases the dispersal of litter and plastic. Crossing highways and roads was also documented in this study, which increases the likelihood of vehicle–animal collisions in the region. Although there is no authentic documentation of human–donkey accidents in the Arabian peninsula and particularly in Saudi Arabia, it is documented in other countries that feral donkeys are responsible for almost 300 human deaths/year [39].

## 5. Conclusions

Our study adds evidence to the detrimental impacts of feral donkeys and calls for urgent actions to control the presence of feral donkeys in the region. These actions include castration, adoption/removal, and eradication [39]. However, castration and removal might not be efficient in controlling the donkeys because castrated individuals can still roam large area and continue to destroy habitats for at least 10 years before they die. Meanwhile, the removal of animals would expand the spatial scale of the donkey, without solving the issue. Humane eradication would be the most efficient and applicable action that can significantly abate the feral donkey’s negative impacts. Humane lethal population control, eradication, is a very common practice and is applied in several parts of the world (Australia, Africa, and America) [40,41], and it was prioritized to control invasive species without conflicting animal welfare standards [42]. Applying a non-poisonous eradication program can have a positive secondary effect on other native species, where it increases food availability for many species, particularly regionally endangered species, such as the Striped hyena (*Hyaena hyaena*) and the Griffon vulture (*Gyps fulvus*) [43,44].

## Figures and Tables

**Figure 1 animals-13-02885-f001:**
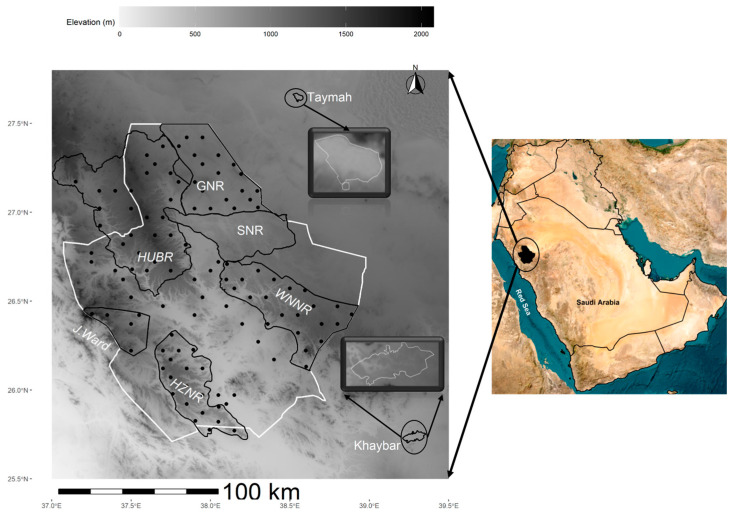
The study area (black area inside the circle in the **right panel**) and the location of the surveyed sites (black dots; *n* = 109) within the boundary (white polygon in the **left panel**) of AlUla region. The black polygons represent the nature reserves and the three areas in the region; abbreviations are defined as per the text.

**Figure 2 animals-13-02885-f002:**
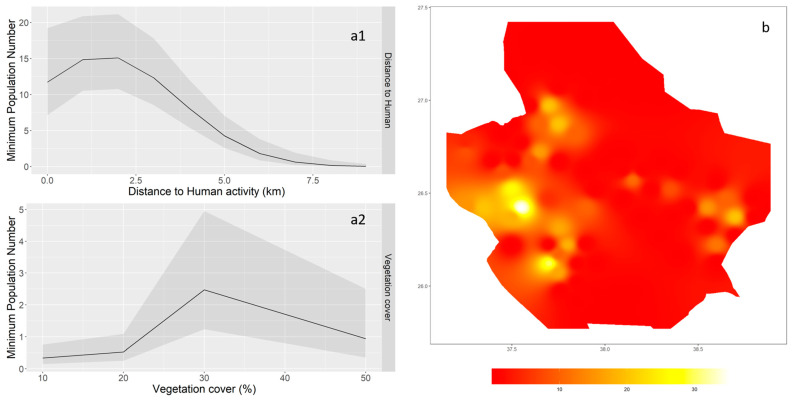
Association between the calculated minimum population number of feral donkeys and the landscape variables on the left (**a1**,**a2**) and the estimated spatial distribution of feral donkeys modeled using Inverse Distance Weighted interpolation (IDW) on the right (**b**).

**Table 1 animals-13-02885-t001:** The minimum population number (MPN) of feral donkeys in the study area.

Site	MPN
Khaybar	338
HUBR	315
Jabal ElWard	235
HZNR	169
WNNR	78
GNR	3
Tayma	0

**Table 2 animals-13-02885-t002:** The model parameters for the best model.

	Estimate	SE	Z Value	*p*-Value
(Intercept)	−1.161	0.419	−2.773	0.0001
Human presence	−2.295	0.488	−4.700	0.0001
Human presence (quadratic term)	−0.671	0.187	−3.581	0.0001
Vegetation cover	2.836	0.546	5.191	0.0001
Vegetation cover (quadratic term)	−0.861	0.198	−4.352	0.0001
Topographic roughness	0.085	0.097	0.867	0.3861

## Data Availability

Primary data not presented in this study are available on request from the corresponding author.

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
