# Peer review of "Feral Donkey Distribution and Ecological Impacts in a Hyper-Arid Region"

_animals, 2023, doi:10.3390/ani13182885_

Round 1

Reviewer 1 Report

Dear authors, 

Many thanks for this piece of work which I read with great interest. I found merit in it and I believe it deserves to be considered further to publication. 

I have very few comments.

The title is informative and concise. Summary is fine. 

Introduction may need some revision. One first point deals with the need for more focused description of donkey morphology. The term feral indicates previous domestication  of breeds. Are those crossbred animals living in the wild now? Mass and size as well as coat colour can be leading factors as to feed selection and space occupancy, including competition and spreading through reproductive success. As to mass and size I would suggest to consider Cappai et al., 2013 IJAS DOI:10.4081/ijas.2013.e29.

Material and methods are well described. Results are informative and discussion is pertinent.

Author Response

We would like to thank the reviewer for the constructive feedback and comments, which have greatly helped to improve our manuscript. 

As per the reviewer’s suggestion, we have now revised the introduction section and provided more details on the morphology of donkeys, and used relevant references. Please refer to lines 39-51 in the revised MS.  

Reviewer 2 Report

This article covers an interesting topic, a definitely understudied feature of arid land ecology.

Below, I list, using line numbers, a few minor suggestions/alternate words and identify aspects of the study that need more explanation.

line 14, insert comma after overgrazing

line 27, insert 'in this part of' northwest Saudi Arabia

line 32, replace urge  with 'propose'

Introduction, general comment: perhaps explain how the reserves in the AlUla region link in with other protected areas, e.g. those of the Saudi Wildlife Authority.

line 81, "three other areas", I think these should be shown more precisely in an expanded Figure 1 (see other comment below)

Section 2.2 Data collection: I accept the justification for the non-random approach to sampling and you describe your 5 x 5 km approach within AlUla county, but you do not explain what you did in the '3 other areas' (Tayma, Khaybar and Jebel ElWard). Did you also follow the 5x5 square method?

line 115, around here, I think you need to add couple of sentences on how you actually worked in a square, e.g. did you aim to drive parallel transects, diagonals etc.? Later you give a distance covered, which indicates  11.6 km driven per  square which would equate with 2 transects.

line 125, vegetation cover; you say you estimated on a 1-100 scale (=%) - how was this estimated across 25 km2? Surely 5 or 10 % 'bins' would be more appropriate.

line 145, 'average density': how was  this calculated, does this apply to AlUla county only or include the other 3 areas?

Figure 2, left hand panel: it is not clear what the x-axes are

line 179, replace almost with 'near'

line 181, lower case for 'Obs'

line 185, state which reserve, GNR?

line 188, please replace existence with 'activity' or 'presence' and likewise for facilities in line 189

line 199, please insert 'Arabian' Oryx, lower case 'g' for gazelle (Idmi or Reem?) and 'Nubian' Ibex.

line 217, Alectoris melanocephala

line 224/5, maybe replace which threatened....'with increased liklihood of vehicle-animal collisions'

line 236, eradication option: if these animals are lethally controlled then perhaps the carcasses could be used to stock 'vulture restaurants' in the reserves. A good outcome for other threatened native taxa in northwestern Saudi Arabia?

line 270, full reference required

The English language is fine, perfectly readable. A few relatively minor suggestions/alternatives are given in the authors comments.

Reviewer 3 Report

This paper describes a survey of feral donkeys in various protected and non-protected areas of northwest Saudi Arabia.  The method of recording donkey numbers and population densities had to be modified and simplified owing the the behaviour of the donkeys upon seeing humans. High numbers of feral donkeys were detected in some areas and few in others.  High numbers were associated with human activities and structures where food resources were more available.  Extensive ecological damage was recorded and the threat of feral donkeys to endangered native ungulates as well as humans through road traffic accidents were discussed.  The authors recommend humane eradication of the donkeys to assist in ecological recovery of affected areas.

Overall this is a very good paper which highlights the high impact that feral donkeys can have on non-native habitats and as such the results are similar to those of other studies in arid regions.  The importance of this paper is to provide data to assist ecological restoration in a large area of northwestern Saudi Arabia, but the lessons of this study could be applied to similar habitats elsewhere.

Please put the word "population" in front of "density".  The density of a donkey means something quite different.

I have made minor comments on the paper.

Overall the quality of the English is very good with only a few  exceptions.  I have made suggestions as to where this can be improved and corrected typos etc.

Round 2

Reviewer 2 Report

I  am satisfied with the prompt responses made by the authors to my comments and feel the manuscript is now in a much better shape and 'fit' for publication.